# Contribution of Geometric Feature Analysis for Deep Learning Classification Algorithms of Urban LiDAR Data

**DOI:** 10.3390/s23177360

**Published:** 2023-08-23

**Authors:** Fayez Tarsha Kurdi, Wijdan Amakhchan, Zahra Gharineiat, Hakim Boulaassal, Omar El Kharki

**Affiliations:** 1School of Surveying and Built Environment, Faculty of Health, Engineering and Sciences, University of Southern Queensland, Springfield Campus, Springfield, QLD 4300, Australia; zahra.gharineiat@usq.edu.au; 2Geomatics, Remote Sensing and Cartography Unit FSTT, Abdelmalek Essaadi University, Tetouan 93000, Morocco; wijdan.amakhchan@etu.uae.ac.ma (W.A.); h.boulaassal@uae.ac.ma (H.B.); elkharki@gmail.com (O.E.K.)

**Keywords:** LiDAR, point cloud, classification, buildings, vegetation, terrain, urban areas, deep learning, machine learning, geometric features

## Abstract

The use of a Machine Learning (ML) classification algorithm to classify airborne urban Light Detection And Ranging (LiDAR) point clouds into main classes such as buildings, terrain, and vegetation has been widely accepted. This paper assesses two strategies to enhance the effectiveness of the Deep Learning (DL) classification algorithm. Two ML classification approaches are developed and compared in this context. These approaches utilize the DL Pipeline Network (DLPN), which is tailored to minimize classification errors and maximize accuracy. The geometric features calculated from a point and its neighborhood are analyzed to select the features that will be used in the input layer of the classification algorithm. To evaluate the contribution of the proposed approach, five point-clouds datasets with different urban typologies and ground topography are employed. These point clouds exhibit variations in point density, accuracy, and the type of aircraft used (drone and plane). This diversity in the tested point clouds enables the assessment of the algorithm’s efficiency. The obtained high classification accuracy between 89% and 98% confirms the efficacy of the developed algorithm. Finally, the results of the adopted algorithm are compared with both rule-based and ML algorithms, providing insights into the positioning of DL classification algorithms among other strategies suggested in the literature.

## 1. Introduction

Airborne Light Detection And Ranging (LiDAR) is a remote sensing system that enables users to scan rural and urban environments with high accuracy, precision, and flexibility. The three-dimensional dataset obtained from LiDAR facilitates the precise mapping and measurement of terrain, building, and vegetation heights, and other features with unparalleled accuracy [1]. By integrating LiDAR sensors onto Unmanned Aerial Vehicles (UAVs), a new tool for LiDAR data generation has emerged. Drone-based LiDAR data exhibit high point density (several hundred points per square meter) and position accuracy (between 5 and 10 cm). Furthermore, this scanning technology is accessible to engineering companies of various scales and can be applied to projects of different sizes and purposes. However, challenges still persist in terms of adhering to UAV regulations, particularly in urban areas and controlled airspaces [2]. Despite the disparities between airborne LiDAR data collected via drones and those obtained through manned aircraft (see Section 3), the efficacy of automated data-processing tools for both types of data requires further investigation.

LiDAR data-processing tools can be categorized based on four main aspects. Firstly, according to their functionality, such as visualization, classification, and modeling tools [3]. Secondly, based on their level of automation, they can be classified as manual, semi-automatic, or fully automatic tools [4]. Thirdly, according to the target class, these tools can focus on a long list of classes such as buildings, terrain, vegetation, roads, powerlines, and railways. Finally, the employed processing algorithms can be grouped into rule-based and Machine Learning algorithms [1].

In recent years, Machine Learning algorithms have emerged as a prominent trend in the automated processing of LiDAR data [5]. Various algorithms, including Random Forest [6], Support Vector Machine (SVM) [7], and Deep Learning (DL) [8], are being utilized in this context. This paper adopts two DL strategies for the automatic classification of airborne LiDAR point clouds in urban areas, focusing on three classes: terrain, vegetation, and buildings. In this context, the DL network consists of three main parts, which are the input layer, hidden layers, and output layer. In fact, the success of the classification network does not depend only on the network structure but also on the used data features in the input layers. The research significance and value of using and calculating the geometric features (GFs) is to help the DL model to be able to distinguish between the classes of point cloud, based on the resulting proprieties of each class. Geometric features can reflect the geometry of the object that the point belongs to when analyzing the relationship between each point in the point cloud and its neighbors. Thus, this paper analyzes the geometric features to select the more efficient ones, to employ them in the input layer of the DL algorithm, which can play a major role in the success of the classification result. Hence, the novel contributions of this paper can be summarized as follows:Analysis and recognition of the relevant geometric features to be employed in Machine Learning (ML) classification algorithms specifically designed for urban areas.Assessment of two classification strategies that achieve highly accurate results.Testing and analysis of the developed classification ML algorithm using data obtained from both unmanned and manned airborne laser scanning.Analysis of the sensitivity of DL classification algorithms across different urban typologies.

The structure of this paper is organized as follows: Section 1 introduces the topic and highlights the novel aspects. Section 2 provides a review of related studies found in the literature. Subsequently, Section 3 presents and analyzes the five datasets used in the study. Section 4, Section 5 and Section 6 delve into the analysis of geometric features, the proposed algorithms, and the discussion of the results obtained. Finally, Section 7 presents the overall research budget and summarizes the main findings.

## 2. Related Works

The state-of-the-art ML classification approaches in the literature for classifying LiDAR data can be categorized into three families based on the level of supervision: unsupervised, weakly supervised or semi-supervised, and supervised.

In the unsupervised ML approach family, Zhang et al. [9] proposed an unsupervised domain-adaptive 3D detection method that utilizes data adaptation from a LiDAR point cloud. Their model was trained by transferring instances from a source domain to target domain scenes through adaptive point distribution. They also introduced an instance-transferring method for selecting and transferring appropriate instances from the source domain to the target domain scene. Furthermore, they employed an adaptive downsampling function to adjust the point-cloud distribution of the transferred instances to approximate the point distribution of the target domain. Similarly, Wang et al. [10] proposed a Nearest-Neighbor-based Contrast Learning Network (NNCNet) as an unsupervised learning algorithm. NNCNet learns discriminative feature representations using large amounts of unlabeled data and incorporates a nearest-neighbor-based data-augmentation scheme to capture accurate inter-modal semantic alignments. The goal of their approach was to learn an encoder that encodes similar data of the same type while maximizing the differences between encoding results of different data classes.

Cortes et al. [11] presented an unsupervised domain-adaptation training strategy for LiDAR point-cloud 3D obstacle detection. Their method utilizes raw points after filtering based on image candidates, and the features encoded by PointNet-like backbones are domain-independent. They significantly improved the accuracy of the model on the target domain without using additional labels by incorporating auxiliary domain classifiers and gradient reversal layers. Zhai et al. [12] adopted a self-supervised approach to enhance the understanding of the point-cloud network by employing a multigrid autoencoder (MA). The MA constrained the coder part of the classification network to gain a deeper understanding of the point cloud during reconstruction. Through self-supervised learning, the performance of the original network also improved.

In the semi-supervised category of the ML approach family, Lei et al. [13] proposed a multi-branch Weakly Supervised Learning Network (WSPointNet) to address the limitations of existing semantic segmentation methods that require a large amount of labeled data, which is labor-intensive and time-consuming. Their method consisted of a basic weakly supervised framework along with a multi-branch weakly supervised module. Yin et al. [14] investigated the potential of weakly supervised semantic segmentation integrated with self-training (WSSS-ST) in handling extensive labeled data training. They introduced a Weakly Supervised Semantic Segmentation (WSSS) framework based on a Semantic Query Net (SQN) to train Deep Neural Networks (DNNs). Their approach aimed to enhance the understanding of 3D scenes, object identification, and the construction of 3D reconstruction models.

Additionally, Li et al. [15] proposed a triplet semi-supervised deep network (TSDN) for the fusion classification of hyperspectral and LiDAR data. They introduced a pseudo-label acquisition strategy to increase the number of available training samples in the semi-supervised learning setting. Their approach also incorporated an attention module to enhance the model’s representation of deep information. The TSDN triplet branch-fusion classification network consisted of a 1-D CNN for spectral feature extraction in hyperspectral imagery (HSI), a 2-D CNN for spatial feature extraction in hyperspectral data, and a Cascade Net for elevation feature extraction in LiDAR data.

In the third family of ML classification approaches, Hassan et al. [16] proposed a deep, hierarchical 3D point-based architecture for object classification and part segmentation. Their architecture captured fine-grained contextual geometric information in the local region, allowing it to learn robust geometric features that are invariant to geometry and orientation. Zhou et al. [17] suggested the Transfer learning-based Sampling Attention Network (TSANet) for the semantic segmentation of 3D urban point clouds, with a focus on developing an effective semantic model for multi-class object detection in urban environments. Xue et al. [18] demonstrated a dynamically sized optimal neighborhood recovery method for 3D point clouds, which considers the correlation between point clouds and the surface variation of local neighborhoods. They extracted local geometric features within the optimal neighborhood to improve the classification accuracy of 3D point clouds. He et al. [19] presented the Stereo Red Green and Blue (RGB) and Deeper LiDAR (SRDL) framework for 3D object detection in autonomous driving, leveraging both semantic and spatial data. They introduce a residual-attention learning mechanism to extract deeper geometric features from irregular 3D point clouds.

Yang et al. [20] adopted a cross-modality feature-fusion network for few-shot 3D point-cloud classification, aiming to recognize objects with only a few labeled samples, even in the presence of missing points in the point-cloud data. They investigated the Support-Query Mutual Attention (SQMA) module for updating support and query features, considering all support features for query feature updating. Chen et al. [21] demonstrated the error feature back-projection-based local-global (EB-LG) feature-learning module to improve point-cloud representation. The EB-LG module facilitated learning hidden features between local and global features, enriching the semantic information of local features. Küçükdemirci et al. [22] suggested a Convolutional Neural Network (CNN) for detecting clearance cairns, which serve as important markers of past agricultural activities. They trained the algorithm using LiDAR-derived datasets that they annotated and created, aiming to develop a specific pre-trained model for the segmentation of specific archaeological features.

Miao et al. [23] proposed Multiscale Feature Fusion (MFFR) based on RandLA-Net [24], which utilized encoder-decoder modules of varying depths to effectively fuse high and low levels of semantic information in point clouds. This technique enhanced the utilization of feature information extracted from the network. Cao et al. [25] presented two semantic segmentation strategies for point clouds: Semantic-Based Local Aggregation (SLA) and Multiscale Global Pyramid (MGP). These strategies addressed the issue of local-feature clustering by leveraging semantic similarity in the local region and incorporating global features in the local feature aggregation. Fan et al. [26] utilized the Multiscale Learning and Attention Enhancement Network (MSLAENet) for the end-to-end classification of HSI and LiDAR data. The network employed a two-branch CNN structure with self-calibrated convolutions and a hierarchical residual structure, enabling the extraction of spectral and spatial information at multiple scales.

Kermarrec et al. [27] compared existing algorithms for the classification of Terrestrial Laser Scanner (TLS) point clouds in landslide monitoring. They analyzed both ML classification algorithms based on manual feature extraction from point clouds and PointNet++ [28], a DL approach that automatically extracts features. Their study aimed to determine the benefits and drawbacks of various classification algorithms. Lei et al. [29] employed a Hierarchical Convolutional Neural Network (H-CNN) model and high-resolution multi-temporal Google Earth images. They explored the combination of hierarchical classification and CNNs to enhance classification accuracy in dense forests. Li et al. proposed the Multi-Level Feature-Extraction Layer (MFEL), which models point clouds at different levels to capture local contextual features and global information. The MFEL extends the GAPLayer [30] and enables feature extraction at three different scales: single point, point neighborhood, and point set, facilitating the extraction of multi-granularity features from point clouds.

Terrain-net, proposed by Li et al. [31], is an end-to-end and highly efficient network that combines the 3D point convolution operator and self-attention mechanism. It effectively captures local and global features for UAV point-cloud ground filtering. This parameter-free DL network is specifically designed for processing UAV point clouds in forested environments. Morsy and Shaker [32] aimed to reduce training data and evaluate the relevance of sixteen LiDAR-derived geometric features for the classification process. They selected the most important features and employed a pointwise classification method based on random forests on a 3D point cloud of a university campus building collected by a TLS system.

Kuprowski and Drozda [33] proposed a single multi-class approach that utilizes a single network to classify all four basic classes (excluding ground) in a power supply domain. They carefully selected features for classifying the airborne LiDAR point cloud, aiming to achieve accurate automatic classification and reduce manual labor. The algorithm utilizes height-based features relative to the ground and requires the creation of a digital terrain model from ground points.

Wang et al. [34] presented a point-cloud processing method based on the Graph Convolution MultiLayer Perceptron (GC-MLP). Their approach improves generalization performance through sparse convolutional links and enhances model capability through dynamic weights and the global receptive field of the MLP. Deng et al. [35] developed a weighted sampling method based on Farthest Point Sampling (FPS) with a DL network that incorporates balanced sampling and hybrid pooling. They adjust the sampling weight values based on the model’s loss to equalize the sampling process and include the relational learning of the sampling center point’s neighborhood space in the feature-encoding process, using a self-attention model to determine feature importance.

Nahhas et al. [36] presented a DL strategy for building detection, and they developed a framework based on an autoencoder to reduce feature dimensionality and a CNN to distinguish between building and non-building objects. Rottensteiner et Briese [37] proposed a method to automatically generate 3D building models from directly observed point clouds collected by LIDAR sensors.

In the context of existing ML- and DL-based methods for point-cloud semantic classification, since PointNet and PointNet++ [28] were introduced in 2017, this has been a very active area of research. Hence, most of the suggested semantic classification approaches focus on the network form to improve their efficiency. This paper focuses on the selection of GFs to enhance the classification results. In turn, the purpose of this work is to evaluate the accuracy of an ML classification strategy for distinguishing the major classes (buildings, terrain, and vegetation) in urban areas. This paper intends to develop and provide a model that highlights the role of geometric features to classify an urban LiDAR point cloud (Figure 1). In this context, our initial focus is on using supervised Machine Learning (ML) techniques commonly used in LiDAR data processing but with a pointed selection of input layer features. Once the target is reached, the next stage will be to create a high-accuracy unsupervised ML classifier.

## 3. Input Data

In this paper, five datasets are used to test the effectiveness of the suggested classification approach explained in Section 5 (Table 1). The selected point clouds represent various urban typologies, with varying point densities and acquisition dates. This choice allows for assessing the reliability of the classification approach across different urban typologies. The first dataset is obtained from Elvis-Elevation and Depth Mapping [38] (see Figure 2). This data sample represents an airborne scan of Brisbane City in Australia (referred to as Brisbane 1 in Table 1). It was acquired between 22 June and 28 October 2014, using a fixed-wing aircraft. The LiDAR sensors used for this scan are ALS50 (Leica, Heerbrugg, Switzerland), SN069, and Riegl 680i (Riegl, Horn, Austria). The creation of the LiDAR dataset involved the utilization of an Inertial Measurements Unit (IMU) and post-processed airborne Global Navigation Satellite System (GNSS) maps in conjunction with the waveform device data. The point cloud captured in this dataset contains various classes, as illustrated in Figure 2, such as building roofs, high and low vegetation, cars, house curbs, sidewalks, swimming pools, utility poles, and sunshades. This paper specifically focuses on extracting three main classes: buildings, high vegetation, and a ground class encompassing other objects.

The second point cloud originates from the southern region of Queensland, Australia, near the city center (refer to Figure 3). It was collected in 2022 using a UAV platform equipped with a DJI M300 RTK and the LiDAR payload Zenmuse L1. The third point cloud is also obtained from Elvis-Elevation and Depth Mapping [38]; this point cloud will be used in Section 6 to analyze the accuracy of the proposed approach. This data sample represents an airborne scan belonging to Brisbane City in Australia. The fourth dataset used in this study pertains to the city of Vaihingen in Germany (see Figure 4), covering an area with small, detached houses and abundant vegetation. Notably, the trees associated with buildings exhibit a significant variation in quality and volume. Lastly, the fifth point cloud represents Christchurch City (see Figure 5) in the Environment Canterbury Regional Council of New Zealand.

Table 1 provides a clear distinction between two types of point density: the mean point density and the theoretical point density. The mean point density is calculated by dividing the number of points within the horizontal area of the scanned zone, while the theoretical point density is provided by the LiDAR provider company. It is worth noting that the mean density should be greater than the theoretical density. This is because the theoretical density does not consider the overlapping areas between scan strips and only considers the beneficial points.

## 4. Geometric Features (GFs)

The geometric features (GFs) are used to study and describe the relationship between each point in the point cloud and its neighbors. Due to the distinct properties resulting from its calculation for each class, the model will be able to easily distinguish between the classes. Indeed, one point and its local neighborhood allow for calculating a list of geometric features [40]. The values of these features express the relative location of these points. In this paper, 11 GFs are tested (Equations (1)–(11) [41]): sum of eigenvalues, omnivariance, eigenentropy, anisotropy, planarity, linearity, PCA1, PCA2, surface variation, sphericity, and verticality. They are based on the eigenvalues $\lambda_{i},i\in \left\{1,2,3\right\}$, eigenvector, and the normal vector.

The sum of eigenvalues is calculated using


(1)
$$\sum \lambda_{i}=\lambda_{1}+\lambda_{2}+\lambda_{3}$$


Anisotropy


(2)
$$A=\frac{\lambda_{1}-\lambda_{3}}{\lambda_{1}}$$


Omnivariance


(3)
$$O=\sqrt[{3}]{\lambda_{1}\times \lambda_{2}\times \lambda_{3}}$$


Eigenentropy


(4)
$$E=-\sum \lambda_{i}\ln(\lambda_{i})$$


Planarity


(5)
$$Ƥ=\frac{\lambda_{2}-\lambda_{3}}{\lambda_{1}}$$


Sphericity


(6)
$$S=\frac{\lambda_{3}}{\lambda_{1}}$$


Surface variation


(7)
$$Sv=\frac{\lambda_{3}}{\lambda_{1}+\lambda_{2}+\lambda_{3}}$$


Linearity


(8)
$$L=\frac{\lambda_{1}-\lambda_{2}}{\lambda_{1}}$$


PCA1


(9)
$$PCA1=\frac{\lambda_{1}}{\lambda_{1}+\lambda_{2}+\lambda_{3}}$$


PCA2


(10)
$$PCA2=\frac{\lambda_{2}}{\lambda_{1}+\lambda_{2}+\lambda_{3}}$$


Verticality


(11)
$$V=1-\left|n_{z}\right|$$


The verticality is derived from the vertical component $n_{z}$ of the normal vector $n\in R^{3}$.

To analyze the various geometric features in the dataset, the Brisbane 1 point cloud is used (see Table 1 and Figure 2). Indeed, the selected point cloud represents a wide area (100 ha), and the terrain topography is not flat. Moreover, different building typologies are present, and the trees are distributed in different forms, e.g., individual trees, groups of connected trees, and trees surrounding buildings. That is why this point cloud can be considered a representative sample. Then, the point cloud is manually classified using CloudCompare software to separate the different classes: building, terrain, and vegetation. Each geometric feature is then calculated for the three main classes. To evaluate the feature values, a histogram graph is generated for each class, representing the list of feature values in the point cloud.

Histograms of calculated anisotropy feature values are shown in Figure 6a–c. The histograms show that most terrain points have distributions with values between 0.9 and 1, while building points have values close to 1 and the vegetation class has distributions with values between 0.75 and 1. This difference in histogram behavior highlights the importance of the anisotropy feature.

The histograms of the eigenentropy feature are shown in Figure 7a–c. In fact, the majority of points have values less than 0.3 for the terrain class, values between 0.7 and 0.8 for the building class, and values between 0.3 and 0.75 for the vegetation class. As a result, the eigenentropy feature can be useful for the separation between the three classes.

The eigenvalues sum is represented as a histogram for each class shown in Figure 8a–c. It can be noted that the value distributions differ significantly between the vegetation, buildings, and terrain classes. This feature is useful to extract terrain classes, but it cannot separate building and vegetation classes.

Figure 9a–c shows the calculated linearity feature values. The histograms show that the distribution of points is nearly identical. Furthermore, despite the histogram spike in variant points, the difference between them is still insignificant, and thus the linearity feature will not aid in the separation of the classes.

Figure 10a–c shows the calculated omnivariance-feature value distributions very clearly between the classes. The terrain points have values between 0.0025 and 0.016, the vegetation points have values between 0.01 and 0.13, and the values for the building points are between 0.03 and 0.1. As a result, this feature cannot aid in distinguishing the three classes.

The calculated PCA1 feature values are shown in Figure 11a–c. The majority of points for each class have a distribution interval between 0.5 and 0.8 or 0.9. These minor differences in the histogram of the PCA1 feature will not support the separation of the vegetation, building, and terrain classes; hence, this geometric feature will be ignored.

The same remake is observed for the PCA2 feature. As a result, the calculated PCA2 values in Figure 12a–c show that the distribution of points in all classes will not benefit us in distinguishing between them.

Figure 13a–c shows histograms of the values derived from the calculated planarity features. Each histogram has a different distribution of points, but they all have similar distribution intervals of between 0 and 1. This feature will not be effective for distinguishing buildings and vegetation from the terrain.

Also, the histograms of the values obtained from the calculated sphericity features are shown in Figure 14a–c. The feature values have different distribution intervals, with the vegetation interval being [0, 0.35], the terrain interval being [0, 0.1], and the building interval being [0, 0.05]. In fact, this feature is important in distinguishing terrain, building, and vegetation classes.

Figure 15a–c shows histograms of the calculated surface-variation feature values. The histograms show the distribution of points at the interval [0, 0.075] for terrain points, for the building class, most points have values at the interval [0, 0.02], whereas vegetation class points have values at the interval [0, 0.15]. Finally, the surface-variation feature can aid in the target classification.

Histograms of the calculated verticality feature values are shown in Figure 16a–c. Most terrain points are located between 0 and 0.05, whereas building points vary from 0 to 0.1 and vegetation class points are between 0 and 1. However, the verticality feature is regarded as efficient.

In summary, the obtained features list will be utilized to assess neighborhood characteristics, shape, and data distribution within the point cloud. These results will facilitate the implementation of the DL pipeline algorithm for recognizing and distinguishing vegetation, building, and terrain points. There are six features that exhibit distinct distributions and values for each class, and they will serve as inputs for the DL pipeline algorithm, along with the coordinates of each point. Table 2 and Figure 17 provide a summary of the geometric features’ scores to test their effectiveness as point-cloud classification criteria.

From Table 2 and Figure 17, it is evident that the input layer of the proposed classification network will incorporate only six geometric features. These features include omnivariance, eigenentropy, sphericity, surface variation, verticality, and anisotropy. After analyzing the geometric features and selecting the relevant ones, the subsequent section will provide a comprehensive description of the proposed classification network.

## 5. Suggested Algorithm

Most of the suggested classification approaches in the literature focus on developing a complex ML classification network for the automatic classification of LiDAR data. The approach suggested in this paper focuses on improving the input layer using basic DL networks. Indeed, it calculates the geometric features of the input point cloud and analyses their values regarding the class that they belong to (see Section 4). Once the efficient geometric features are recognized and adopted in the input layer, the training stage can start to calculate the trained classification model.

In this section, the proposed algorithm primarily focuses on classifying the point cloud into four major classes: buildings, vegetation, ground, and noise. It is important to note that the noise class may encompass various unconsidered objects typically found in urban environments, such as cars, house curbs, sidewalks, swimming pools, utility poles, low vegetation, powerlines, and traffic lights. Grouping these objects into a single noise class is a deliberate choice, as the three considered classes collectively account for more than 85% of the analyzed urban point clouds (refer to Table 3). This remake highlights the importance of the three considered classes. The decision is taken to consider the buildings, vegetation, and terrain as the main three components of an urban scene point cloud, and that all other points can be classified within one class currently named “noise class”. Future research will focus on the automatic extraction of different minor classes from noise class regarding their importance in the final 3D city model.

### 5.1. Classification Network

The input layer is the labeled point cloud that contains the coordinates and the calculated features of each point. The hidden layers of the MLP classifier can be one hidden layer or more. The output layer is the result obtained by the hidden layer’s strict calculations. Figure 18 demonstrates the process and how the DL works.

The activation function employed is the logistic function (Equation (13)), where HL is the hidden layers. In this paper, ten HLs are used. m is the number of dimensions equal to nine (XYZ coordinates plus six selected features—see the GFs section). The parameters and hyperparameters in a neural network can affect the model’s performance. The main parameters are weights, biases, learning rate, number of hidden layers, and the activation function. Each node has a certain amount of weight that is assigned to it in the network. The weighted sum of the inputs and the bias is calculated using Equation (12) [42].
(12)$$S=b+\sum_{k=1}^{m}x_{ik}w_{ik}$$
where $S\left(.\right):R^{m}\to R^{o}$, m is the number of input dimensions, and o is the number of output dimensions [43,44]. $X=[x_{1},\;x_{2},\;\ldots ,\;x_{m}]$ is the input features, $W=[w_{1},\;w_{2},\;\ldots ,\;w_{m}]$ their weights, and b is a bias term [45].

Hence, the activation function receives the results of the calculated sum S, and it returns the accurate result from the node based on the output received (see Equation (13) which represents the logistic sigmoid activation function [46]).
(13)$$\varphi \left(S\right)=logistic\left(S\right)=\frac{1}{{1+e}^{-S}}$$

The selection of activation function controls how perfectly a network model works to learn the training dataset while in the output layer, and it determines the types of predictions a model can generate. Once the final output is obtained, the loss function is employed to compute the difference between the predicted and the resulting outputs; it measures how well the neural network models the training data. The loss function employed by MLP is the log-loss calculated using Equation (14) [47].
(14)$$L\left(P,Y\right)=-\frac{1}{N}\sum_{i=0}^{N}\sum_{k=0}^{K}y_{i,k}{log(p}_{i,k})$$
where N is the number of samples and K is the number of labels, $Y={[y}_{1},\;y_{2},\;\ldots ,\;y_{N}]$ matrix of true labels, $y_{i,k}=1$ if sample i has label k taken from a set of K labels, $P={[p}_{1},\;p_{2},\;\ldots ,\;p_{N}]$ is a matrix of probability estimated, with $p_{i,k}=\mathrm{Pr}\left(y_{i,k}=1\right)$.

To calculate the percentage of correct predictions made by a given model, the accuracy score is used. In ML applications, the accuracy score is an evaluation metric that compares the number of correct predictions made by a model to the total number of predictions made. It is computed by dividing the total number of predictions by the number of correct predictions using Equation (15) [47].
(15)$$Accuracy\left(\hat{y_{i}},y_{i}\right)=\frac{Number\;of\;correct\;predictions}{Total\;number\;of\;predictions}=\frac{1}{m}\sum_{i=0}^{m-1}1\left(\hat{y_{i}}=y_{i}\right)$$
where $\hat{y_{i}}$ is the predicted value of the ith sample and $y_{i}$ is the corresponding true value, then the fraction of correct predictions over m is the number of input dimensions.

### 5.2. Method

To achieve LiDAR point-cloud classification, this paper proposes two strategies. The first strategy involves applying the proposed algorithm to the point cloud, extracting all four classes simultaneously. The second strategy follows a two-step classification process. Firstly, the point cloud is classified into buildings and non-buildings, and then the non-building class is further classified into vegetation, terrain, and noise classes based on the features calculated in the geometric feature (GF) section.

The primary focus of this study is the classification of the urban scene into three classes: terrain, vegetation, and buildings, making the first scenario the direct objective. The analysis of geometric features indicates that the selected features effectively distinguish the building mask within the examined point cloud. Furthermore, the accurate separation of vegetation and terrain classes has been supported in the existing literature [8]. These two reasons serve as the main motivations for developing the second scenario. The distinction between the two methods lies in the labeling of the input data. In the first scenario, the point cloud is labeled into four classes. In contrast, the second scenario involves a labeling process where the model is initially assigned two classes: buildings are labeled as 2 and non-buildings are labeled as 0. Subsequently, the non-building class is further labeled as vegetation (1) and non-vegetation (0), and finally, the non-vegetation is labeled as ground (0) and non-ground (3). Consequently, the point cloud is divided into four classes as a result of this process.

The subsequent section will present the results of applying the two strategies to the LiDAR point cloud and evaluate the classification accuracy in both cases.

### 5.3. DLPA Workflow

The suggested algorithm begins by performing downsampling on the point cloud, which involves selecting a subset of samples to reduce the sampling rate. This process helps reduce data size and density, making it suitable for ML algorithms that require fewer training points. Downsampling also allows for the creation of smaller models, freeing up device space for other necessary codes and processes.

The second step involves the manual labeling of the point cloud using software tools such as CloudCompare. In this research, CloudCompare software is utilized to manually classify the LiDAR data. The classification process starts by extracting buildings, vegetation, and ground points separately. Any remaining points that do not belong to the considered classes are categorized as noise. Each class is assigned a specific code: 0 for ground points, 1 for vegetation points, 2 for building points, and 3 for the noise class.

The third step involves calculating the selected features from the GF section for each class. The features include omnivariance, eigenentropy, sphericity, surface variation, verticality, and anisotropy. Histograms are then generated to analyze the values obtained from each feature for the four classes. This analysis helps to identify the most accurate features that can effectively differentiate between the classes. Subsequently, the calculated features are combined with the labeled point cloud to create the input data for the DLPA.

The input point cloud is divided into three sets: 70% for the training set, 10% for the validation set, and 20% for the test set. The DLPA is implemented using Python, NumPy, and the MLP classifier algorithm from the Sklearn libraries [43]. Thereafter, the DLPA is trained using the training dataset. The trained DLPA is then validated using the validation dataset to assess its performance and effectiveness. Finally, the test dataset is utilized to further validate the trained and validated DLPA by adjusting the hyperparameters. The workflow of the DLPA is summarized in Figure 19.

The output of the DLPA is the prediction label for each point of the city point cloud, e.g., 2 for building class points, 1 for the vegetation class, 0 for the terrain class, and 3 for the noise class. The obtained classification results will be mentioned in the Results section.

To improve the obtained results, the one-vs-the-rest (OvR) multi-class strategy is used [43], also known as one-vs-all. This strategy consists of fitting one classifier per class, which takes the classifier as a parameter, which is in this case the DLPA. For each classifier, the class is fitted against all the other classes. The hyperparameters that are used to obtain the best accuracy results are 10 hidden layers, the Adam solver, and the logistic sigmoid activation function.

## 6. Results and Accuracy

The objective of the suggested approach is to extract the classes of buildings, vegetation, and terrain from an urban point cloud using the DLPA, as illustrated in Figure 19. This algorithm is applied using two different strategies. The first strategy extracts the three target classes simultaneously, while the second strategy performs the classification in two consecutive steps: the classification of buildings/non-buildings and the classification of the non-buildings class into three classes: terrain, vegetation, and noise. Figure 20a, Figure 21a, Figure 22a and Figure 23a present the results obtained by applying the suggested approach using the first strategy, while Figure 20b, Figure 21b, Figure 22b and Figure 23b depict the results obtained using the second strategy. These figures clearly demonstrate the effective completion of building, vegetation, and terrain classification in both scenarios. In fact, the developed approach achieves an accuracy score of 97.9% when using the first method and 98.8% when using the second. The accuracy is calculated by comparing the results to a manually classified point cloud serving as the reference model.

It is important to highlight that the first scenario not only offers a simpler approach compared to the second scenario but it also requires applying the DLPA only once. As a result, it demands less processing time compared to the second classification strategy. However, it should be noted that the second scenario achieves a slightly higher accuracy score of 98.8% compared to the 97.9% achieved in the first scenario. The results of the second point cloud are presented in Figure 24, Figure 25 and Figure 26.

The third point cloud is the Vaihingen point cloud, the results are shown in Figure 27, Figure 28 and Figure 29.

The results of the Christchurch point cloud are in Figure 30, Figure 31 and Figure 32.

The results can also be represented by categories, e.g., building, vegetation and terrain as showed in Figure 33, Figure 34, Figure 35 and Figure 36.

The first scenario consists of applying the proposed algorithm to the point cloud and extracting the three classes at the same time, aiming to classify the entire point cloud into buildings, vegetation, and terrain without intermediate steps. This approach offers a straightforward implementation by processing the entire point cloud in a single step, eliminating the complexity of dividing the process into multiple stages. Additionally, class coherence is enhanced as all four classes are classified together, allowing the algorithm to consider relationships and interactions between classes for improved accuracy. While processing the entire point cloud at once may pose computational challenges, efficient algorithms and hardware can ensure acceptable performance. With a high accuracy score of 96.9%, this strategy demonstrates its effectiveness in accurately identifying the different classes within the point cloud.

On the other hand, the second strategy adopts a two-step approach for classifying the point cloud. The initial step focuses on distinguishing buildings from non-building objects, while the second step classifies the non-building objects into three categories: vegetation, terrain, and noise, achieving a score of 98%. This strategy’s advantage lies in its ability to prioritize the accurate identification of buildings, which are often of utmost importance in various applications. By separating the classification into two stages, the algorithm can specifically adjust the feature-extraction and classification techniques for each step, potentially improving accuracy. Additionally, the division of non-building objects into three distinct classes allows for a more detailed and precise analysis. This strategy offers flexibility for the customization and fine-tuning of algorithms for each class, allowing for tailored approaches. However, there is a potential risk of information loss during the intermediate classification stage, although this can be mitigated by careful feature and algorithm selection.

Both strategies rely on the calculation of geometric features to achieve efficient results. Geometric features provide valuable information about the shape, size, and spatial relationships of objects within the point cloud. By leveraging these features, both strategies can effectively differentiate between different classes. The use of geometric features allows the algorithms to capture important characteristics unique to each class and enable accurate classification.

The performance of the model is tested using the Brisbane 3 dataset (Figure 37), where the trained model of the Brisbane 1 dataset is applied to the Brisbane 3 dataset. The obtained results are represented in Figure 38, Figure 39, Figure 40, Figure 41 and Figure 42, with an accuracy score of 89%. Despite the high similarity in typology and point-cloud characteristics between the trained and tested data, the accuracy of the obtained result is still lower than in the case of using the same dataset (98%). However, the obtained accuracy still represents a promising result. More investigation in future work will be conducted to improve the transportation accuracy of the trained model parameters for other datasets. Figure 37 mentions the visualization of the Brisbane 3 point cloud, whereas Figure 38, Figure 39, Figure 40, Figure 41 and Figure 42 illustrate the classification results.

At this stage, it is important to underline numerous authors who investigated the extraction of objects from urban point clouds using LiDAR data and using rule-based approaches. In one such study, Demir [48] uses the RANdom SAmple Consensus for (RANSAC) paradigm [49] to recognize buildings in urban LiDAR point clouds. The resulting accuracy of this algorithm ranges between 77% and 92%. In the same context, Shao et al. [50] adopt a top-down rule-based algorithm to detect seed-point sets with semantic information; then, a region-growing algorithm is applied to detect the building class. The resulting accuracy of this algorithm ranges between 0.65% and 98.9%. It can be noted that the building-extraction accuracy in both approaches varies according to the employed accuracy factor (correctness, completeness, quality, and error types 1 and 2), but in all cases, the obtained accuracy of the suggested DL algorithm is still promising (98%). In the same context, Table 4 and Figure 43 show a comparison between the second-used classification strategy and three rule-based approaches that participated in the project of “ISPRS benchmark on urban object detection” [51,52,53,54]. All approaches in this comparison use the Vaihingen dataset (Table 1 and Figure 4). The building detection accuracy in the cities’ approaches was estimated using correctness, completeness, and quality as accuracy factors. The mentioned results in this comparison confirm the validity of the geometric features analysis in the context of DL LiDAR data classification.

Finally, it is useful to calculate the accuracy of various classes among first and second classification strategies (Table 5). It can be noticed that the maximum difference between obtained accuracy equals 3%, and the building class has the highest extraction accuracy (99%) in the second scenario.

## 7. Conclusions

This paper assessed a DL pipeline algorithm, based on the MLP Neural Network, for the automatic classification of urban LiDAR point clouds. The algorithm utilized both the point coordinates and the geometric features of the LiDAR points, which have been analyzed to identify the most effective features for distinguishing between different objects in the point cloud. The ML algorithm parameters have also been optimized to achieve optimal performance.

Two classification strategies are employed in this study. The first strategy directly classifies the urban point cloud into terrain, vegetation, and buildings. The second strategy initially classifies the point cloud into buildings and non-buildings, followed by the classification of the non-building class into vegetation and non-vegetation. Lastly, the non-vegetation class is further classified into terrain and non-terrain.

Five LiDAR point clouds from various areas were used to evaluate the proposed classification approach. The achieved high classification accuracy of 98% validates the efficacy of the algorithm. It is important to note that the second strategy requires more processing time compared to the first strategy, but it yields slightly more accurate results.

In conclusion, the obtained accuracy results are highly promising. Further research is needed to investigate the transferability of the trained model to similar data, which would provide an advantage for supervised ML classification over unsupervised ML. Nonetheless, unsupervised ML remains a valuable research option that can contribute significantly to LiDAR data classification. Additionally, it is recommended to test the proposed approach on datasets representing different urban typologies to further validate its effectiveness. Finally, generalizing the solution and applying it to other point clouds with varying scales, orientations, scanning resolution, and urban scenery is the next step in using the model directly to any point cloud.

## Figures and Tables

**Figure 1 sensors-23-07360-f001:**
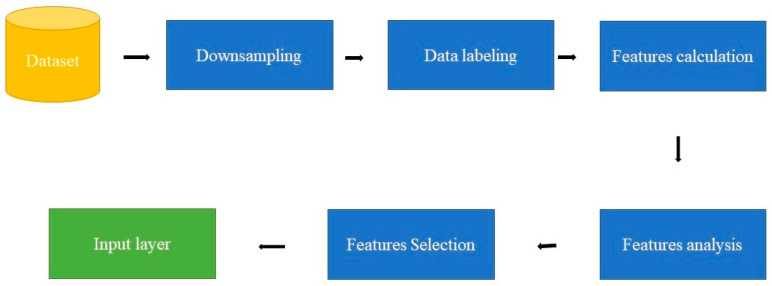
General phases to generate the input layer.

**Figure 2 sensors-23-07360-f002:**
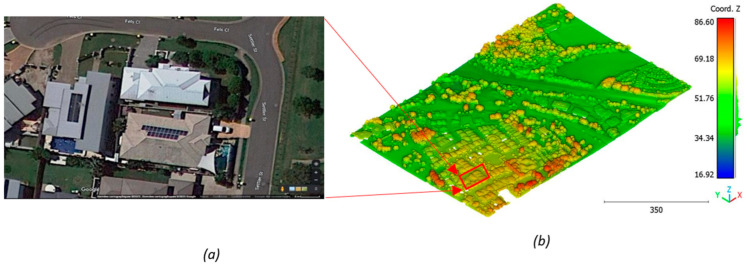
Point cloud of Brisbane city (Brisbane 1); (**a**) Google Maps image; (**b**) LiDAR point cloud.

**Figure 3 sensors-23-07360-f003:**
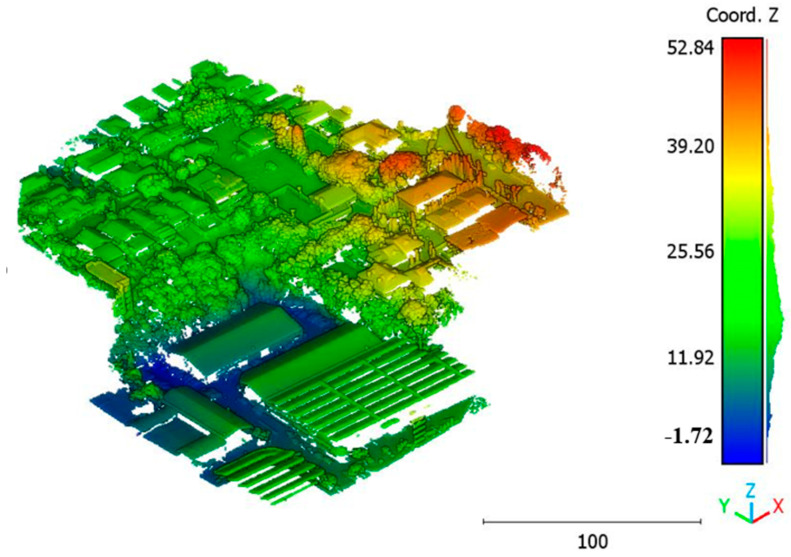
Point cloud of south area of Queensland, Australia (Brisbane 2).

**Figure 4 sensors-23-07360-f004:**
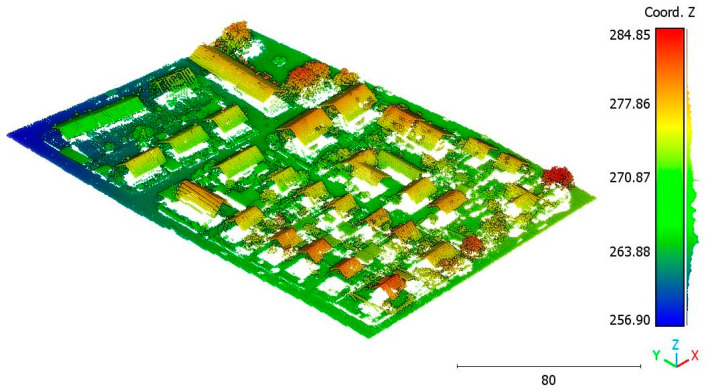
Vaihingen city (Germany) point cloud.

**Figure 5 sensors-23-07360-f005:**
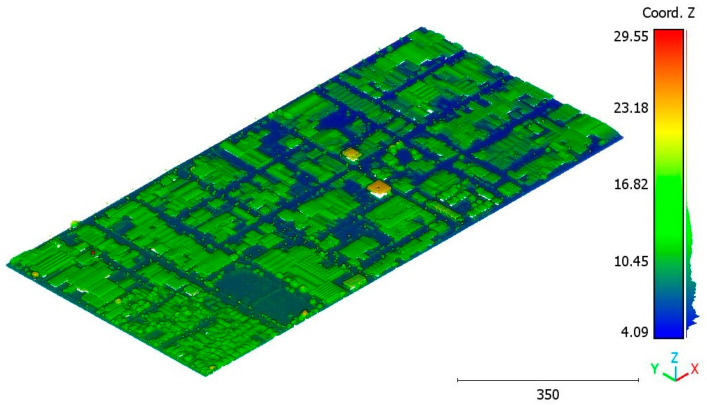
Christchurch city point cloud.

**Figure 6 sensors-23-07360-f006:**
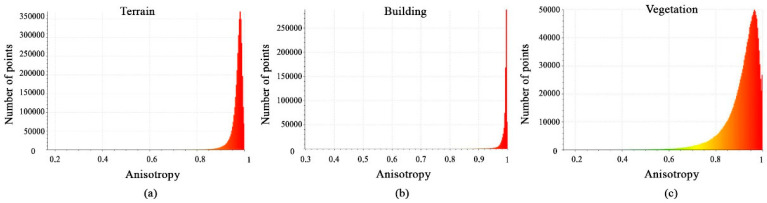
Histograms of different values of anisotropy feature. (**a**) Terrain class; (**b**) Building class; (**c**) Vegetation class.

**Figure 7 sensors-23-07360-f007:**
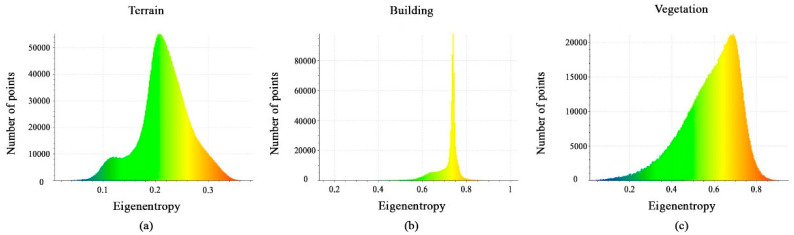
Histograms of different values of the eigenentropy feature. (**a**) Terrain class; (**b**) Building class; (**c**) Vegetation class.

**Figure 8 sensors-23-07360-f008:**
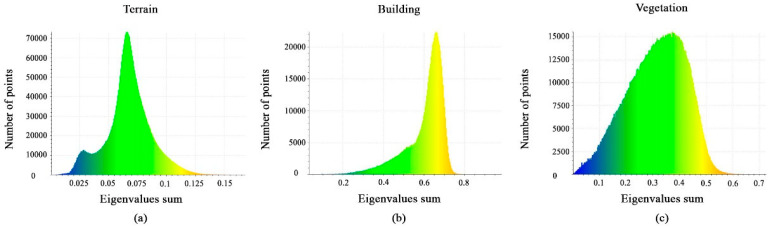
Histograms of different values of eigenvalues sum feature. (**a**) Terrain class; (**b**) Building class; (**c**) Vegetation class.

**Figure 9 sensors-23-07360-f009:**
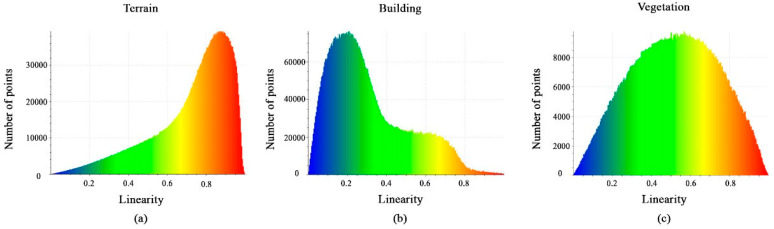
Histograms of different values of linearity feature. (**a**) Terrain class; (**b**) Building class; (**c**) Vegetation class.

**Figure 10 sensors-23-07360-f010:**
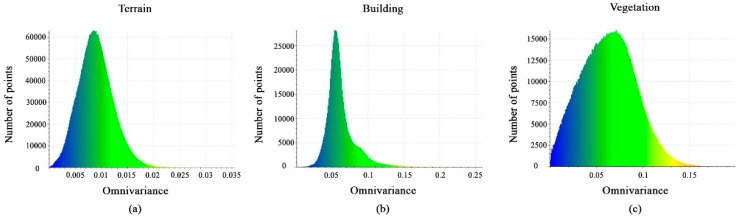
Histograms of different values of the omnivariance feature. (**a**) Terrain class; (**b**) Building class; (**c**) Vegetation class.

**Figure 11 sensors-23-07360-f011:**
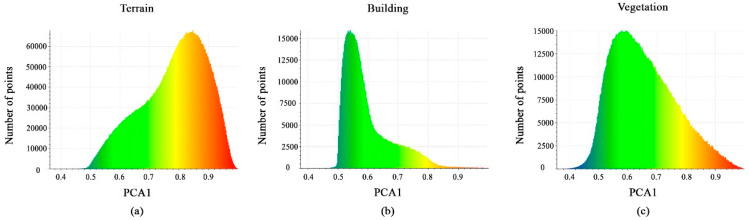
Histograms of different values of PCA1 feature. (**a**) Terrain class; (**b**) Building class; (**c**) Vegetation class.

**Figure 12 sensors-23-07360-f012:**
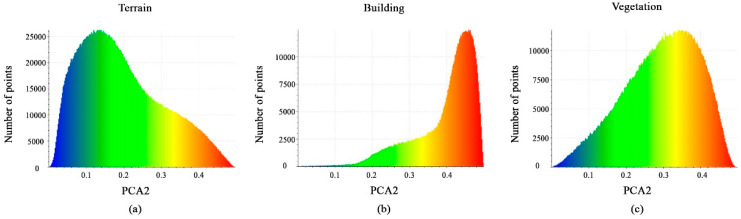
Histograms of different values of PCA2 feature. (**a**) Terrain class; (**b**) Building class; (**c**) Vegetation class.

**Figure 13 sensors-23-07360-f013:**
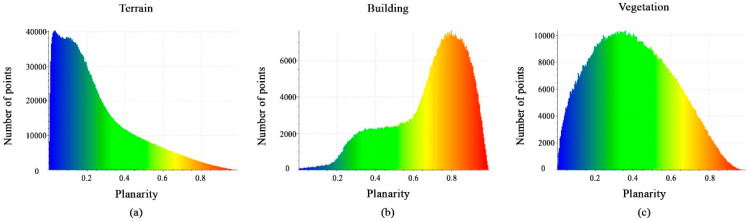
Histograms of different values of planarity feature. (**a**) Terrain class; (**b**) Building class; (**c**) Vegetation class.

**Figure 14 sensors-23-07360-f014:**
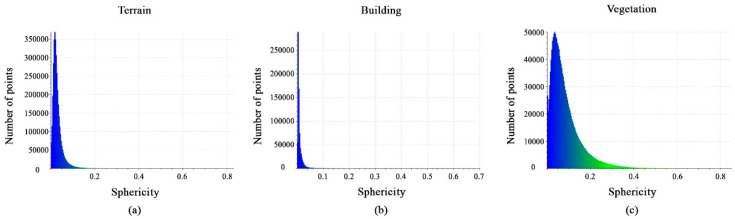
Histograms of different values of sphericity feature. (**a**) Terrain class; (**b**) Building class; (**c**) Vegetation class.

**Figure 15 sensors-23-07360-f015:**
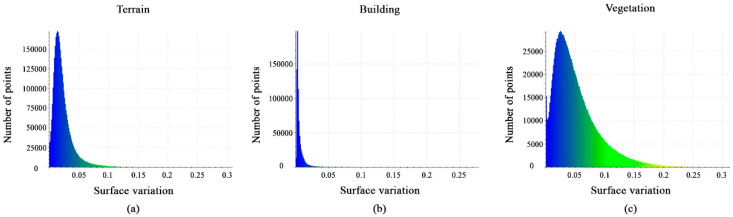
Histograms of different values of surface-variation feature. (**a**) Terrain class; (**b**) Building class; (**c**) Vegetation class.

**Figure 16 sensors-23-07360-f016:**
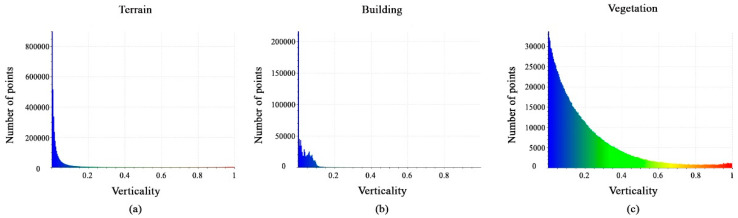
Histograms of different values of verticality feature. (**a**) Terrain class; (**b**) Building class; (**c**) Vegetation class.

**Figure 17 sensors-23-07360-f017:**
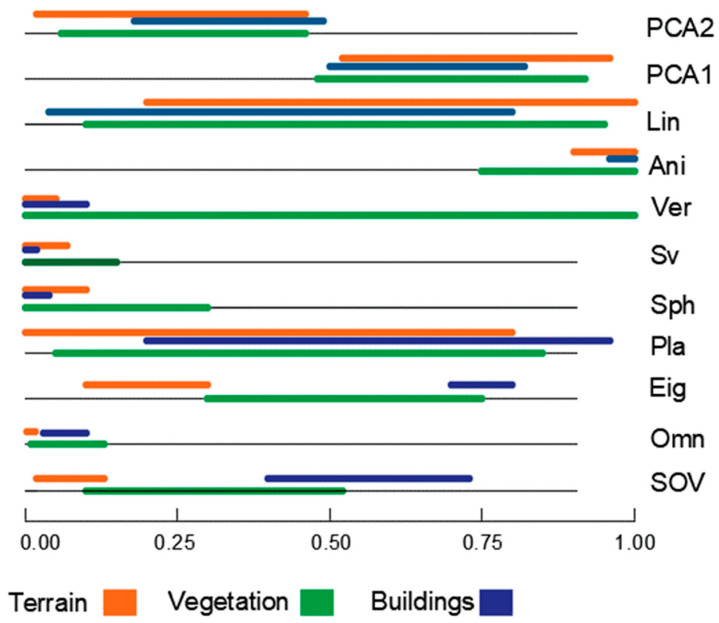
Graphical representation of geometric features.

**Figure 18 sensors-23-07360-f018:**
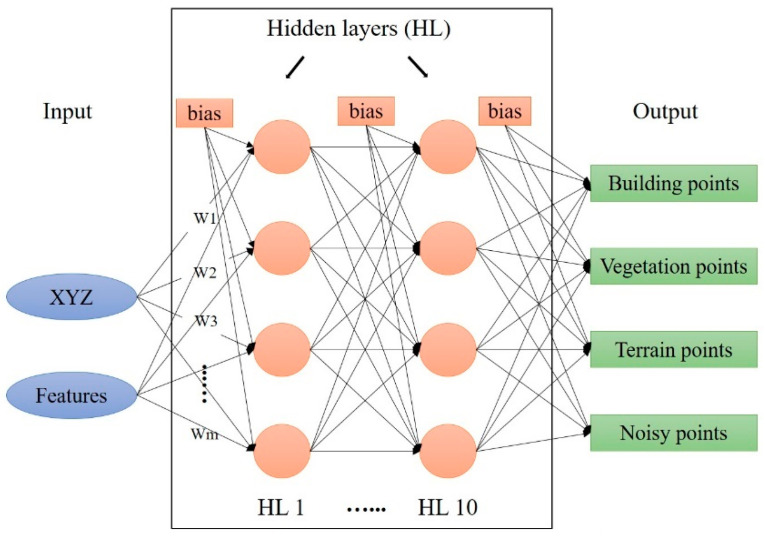
DL model architecture.

**Figure 19 sensors-23-07360-f019:**
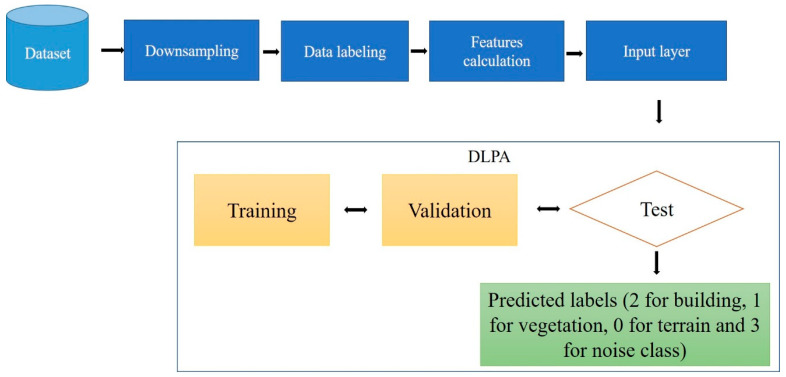
DL pipeline algorithm workflow.

**Figure 20 sensors-23-07360-f020:**
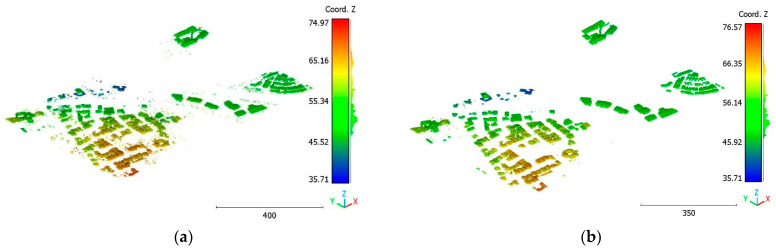
Predicted building class of Brisbane 1 point cloud using (**a**) First scenario. (**b**) Second scenario.

**Figure 21 sensors-23-07360-f021:**
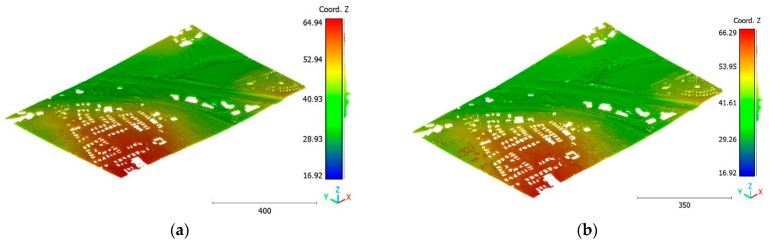
Predicted terrain class of Brisbane 1 point cloud using (**a**) First scenario. (**b**) Second scenario.

**Figure 22 sensors-23-07360-f022:**
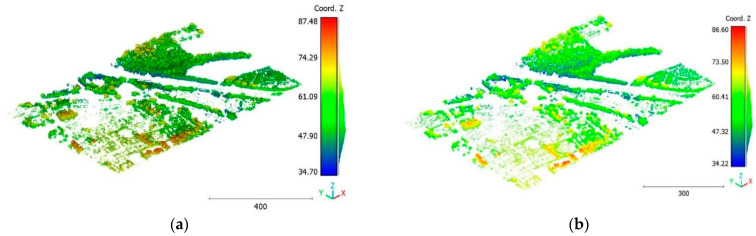
Predicted vegetation class of Brisbane 1 point cloud using (**a**) First scenario. (**b**) Second scenario.

**Figure 23 sensors-23-07360-f023:**
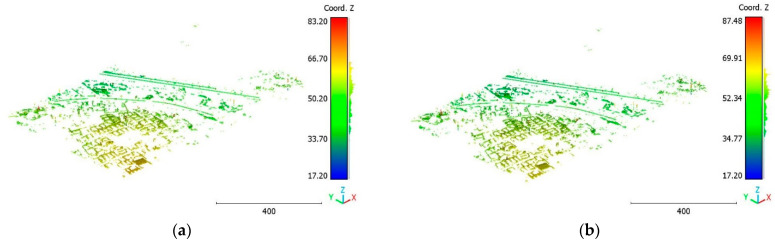
Predicted noise class of Brisbane 1 point cloud using (**a**) First scenario. (**b**) Second scenario.

**Figure 24 sensors-23-07360-f024:**
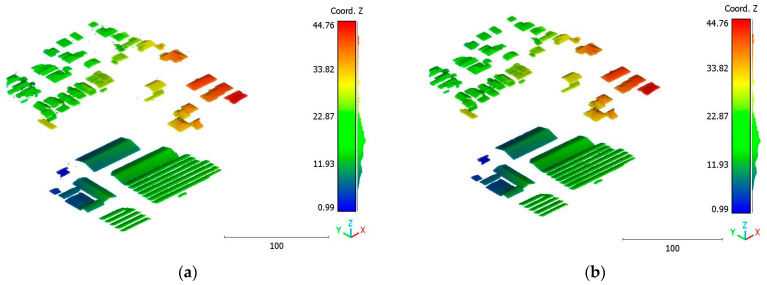
Predicted building class of south area of Queensland, Australia, point cloud using (**a**) First scenario. (**b**) Second scenario.

**Figure 25 sensors-23-07360-f025:**
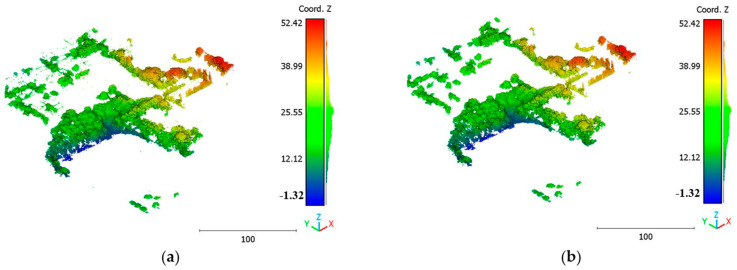
Predicted vegetation class of south area of Queensland, Australia, point cloud using (**a**) First scenario. (**b**) Second scenario.

**Figure 26 sensors-23-07360-f026:**
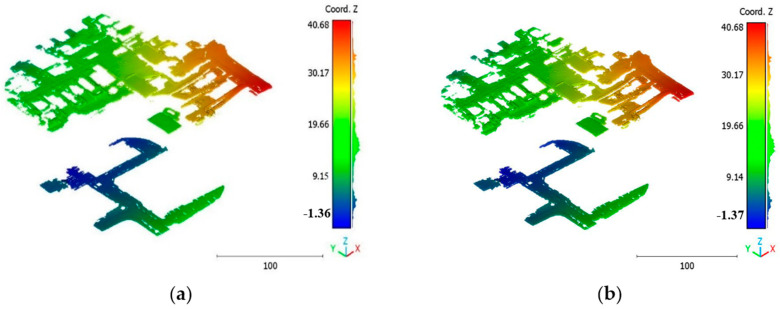
Predicted terrain class of south area of Queensland, Australia, point cloud using (**a**) First scenario. (**b**) Second scenario.

**Figure 27 sensors-23-07360-f027:**
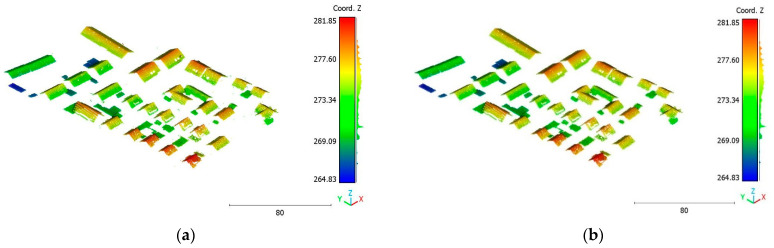
Predicted building class of Vaihingen point cloud using (**a**) First scenario. (**b**) Second scenario.

**Figure 28 sensors-23-07360-f028:**
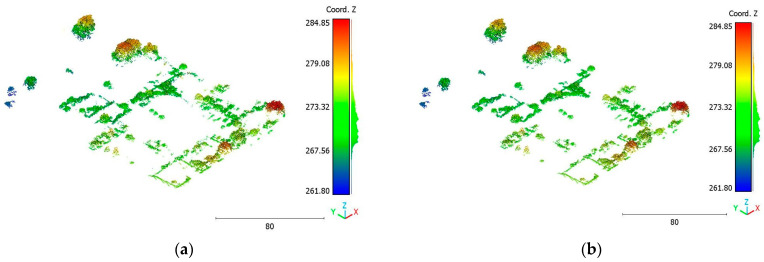
Predicted vegetation class of Vaihingen point cloud using (**a**) First scenario. (**b**) Second scenario.

**Figure 29 sensors-23-07360-f029:**
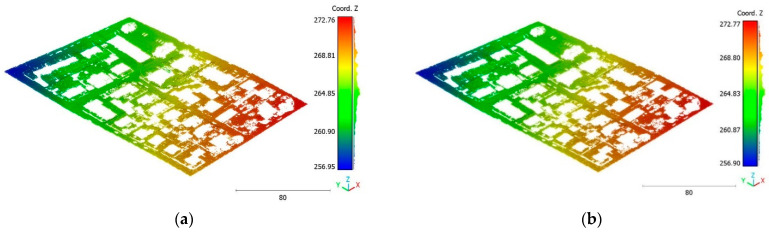
Predicted terrain class of Vaihingen point cloud using (**a**) First scenario. (**b**) Second scenario.

**Figure 30 sensors-23-07360-f030:**
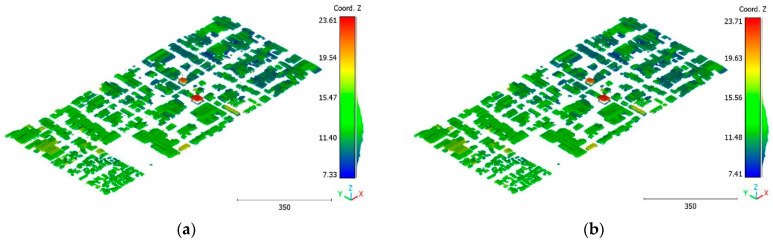
Predicted building class of Christchurch point cloud using (**a**) First scenario. (**b**) Second scenario.

**Figure 31 sensors-23-07360-f031:**
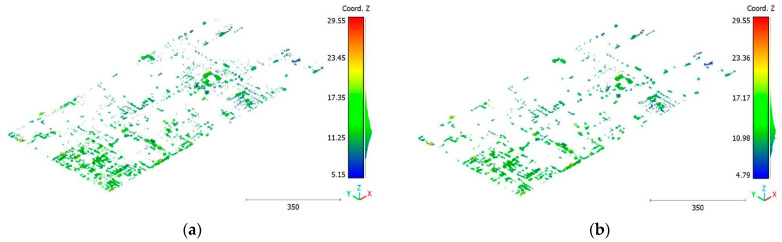
Predicted vegetation class of Christchurch point cloud using (**a**) First scenario. (**b**) Second scenario.

**Figure 32 sensors-23-07360-f032:**
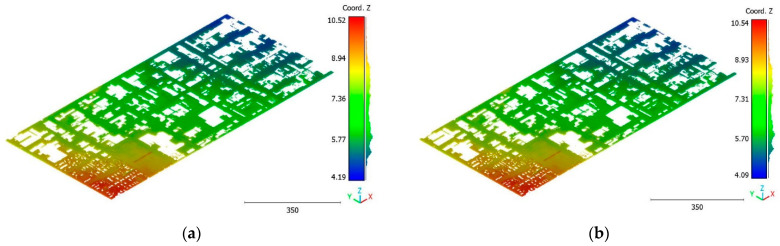
Predicted terrain class of Christchurch point cloud using (**a**) First scenario. (**b**) Second scenario.

**Figure 33 sensors-23-07360-f033:**
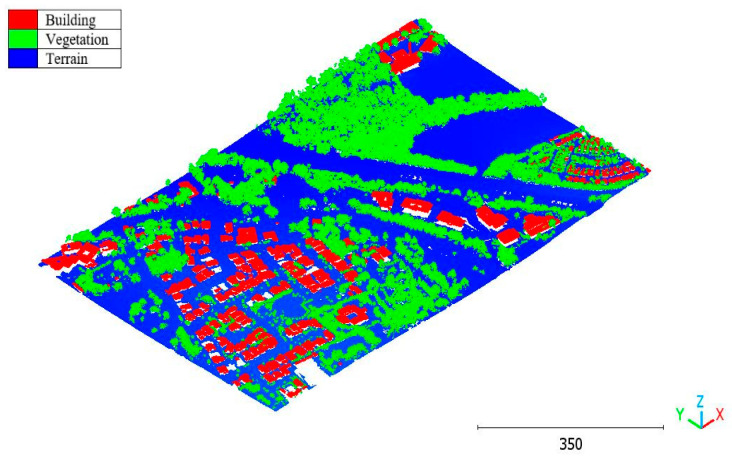
Brisbane 1 point cloud by categories.

**Figure 34 sensors-23-07360-f034:**
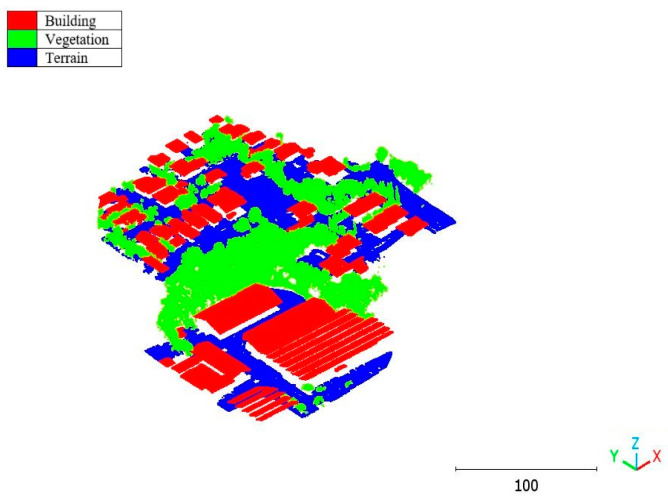
South area of Queensland Australia point cloud by categories.

**Figure 35 sensors-23-07360-f035:**
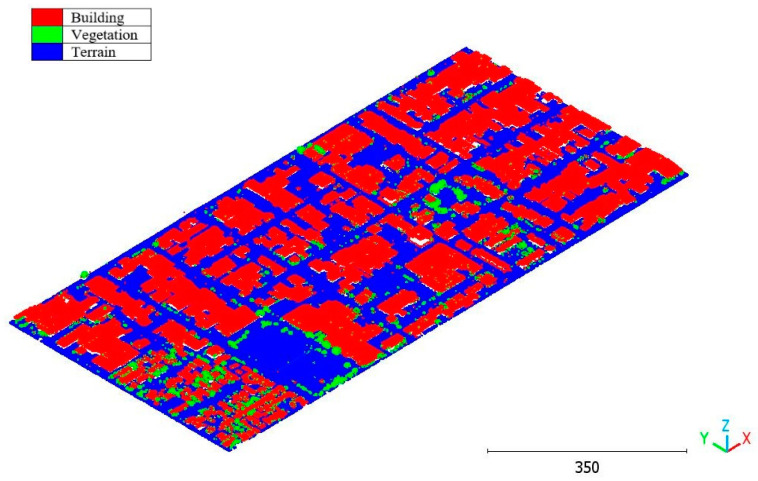
Christchurch point cloud by categories.

**Figure 36 sensors-23-07360-f036:**
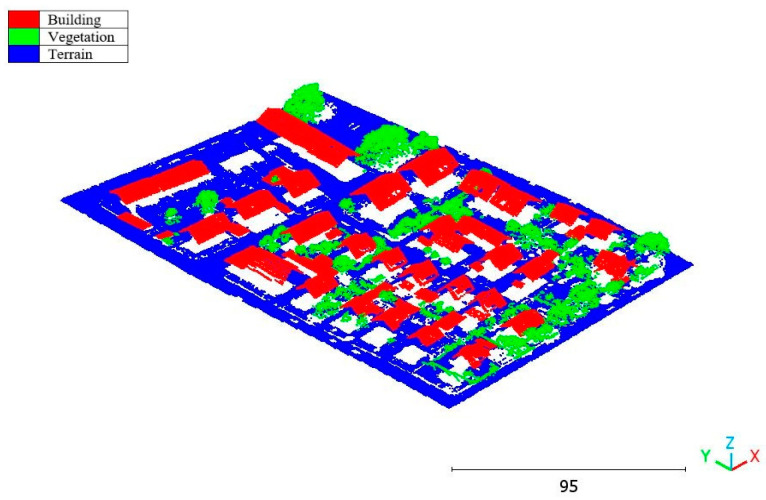
Vaihingen point cloud by categories.

**Figure 37 sensors-23-07360-f037:**
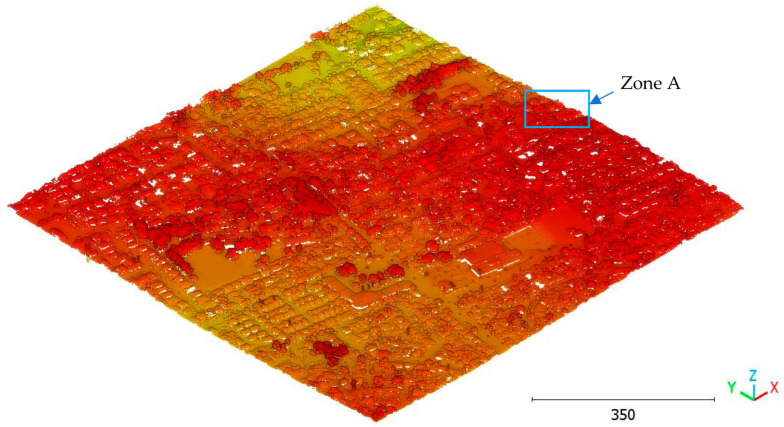
Point cloud of Brisbane city (Brisbane 3).

**Figure 38 sensors-23-07360-f038:**
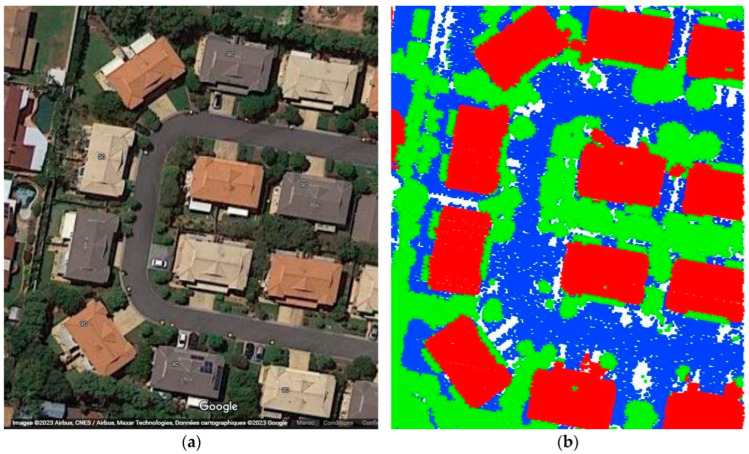
Zone A of Brisbane city; (**a**) Google Maps image; (**b**) LiDAR point cloud obtained by categories; Red is buildings, green is vegetation, blue is terrain.

**Figure 39 sensors-23-07360-f039:**
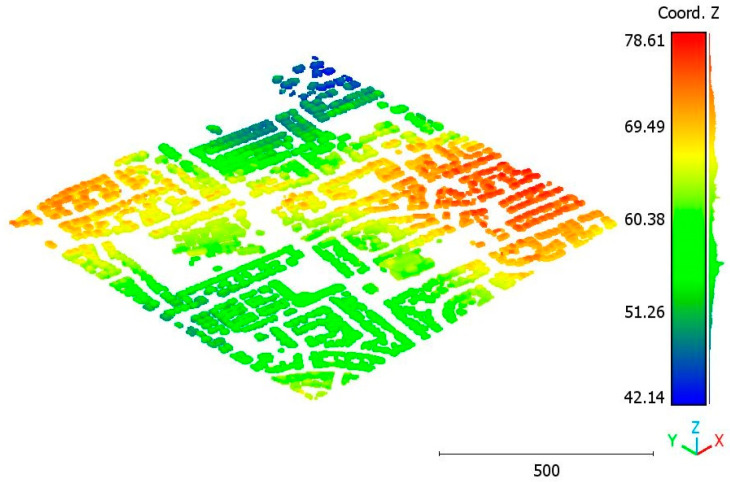
Predicted building class of Brisbane 3 point cloud.

**Figure 40 sensors-23-07360-f040:**
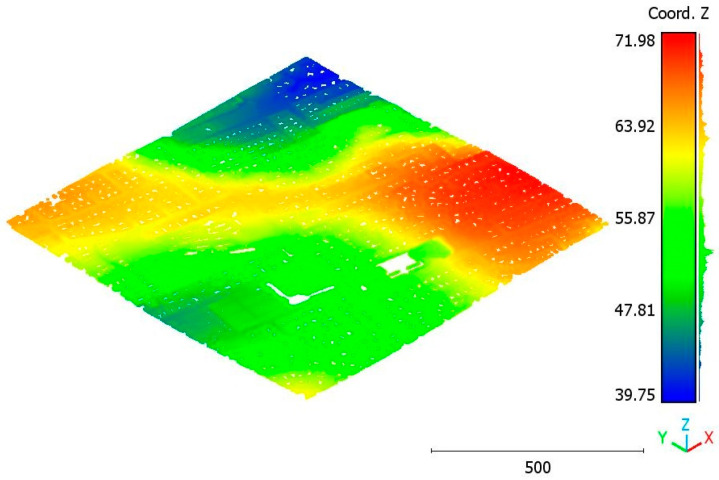
Predicted terrain class of Brisbane 3 point cloud.

**Figure 41 sensors-23-07360-f041:**
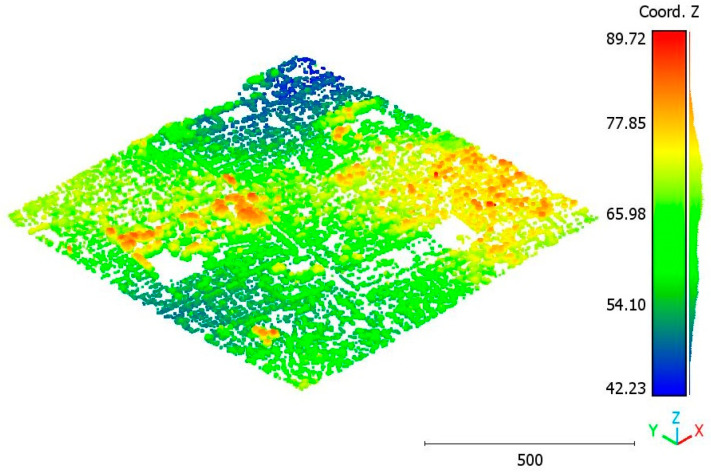
Predicted vegetation class of Brisbane 3 point cloud.

**Figure 42 sensors-23-07360-f042:**
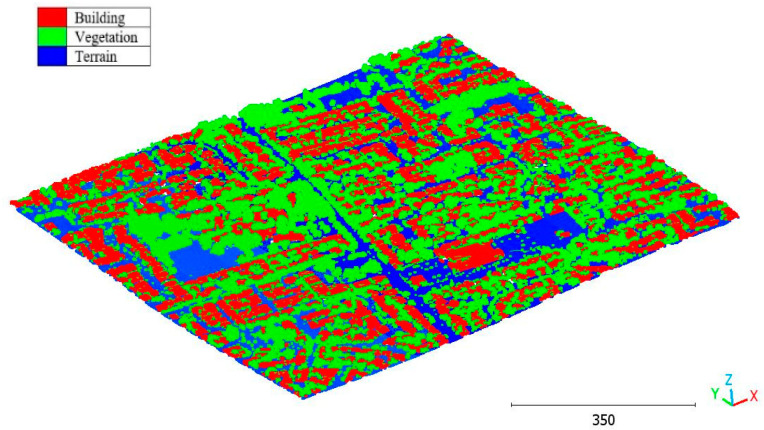
Brisbane 3 point cloud by categories.

**Figure 43 sensors-23-07360-f043:**
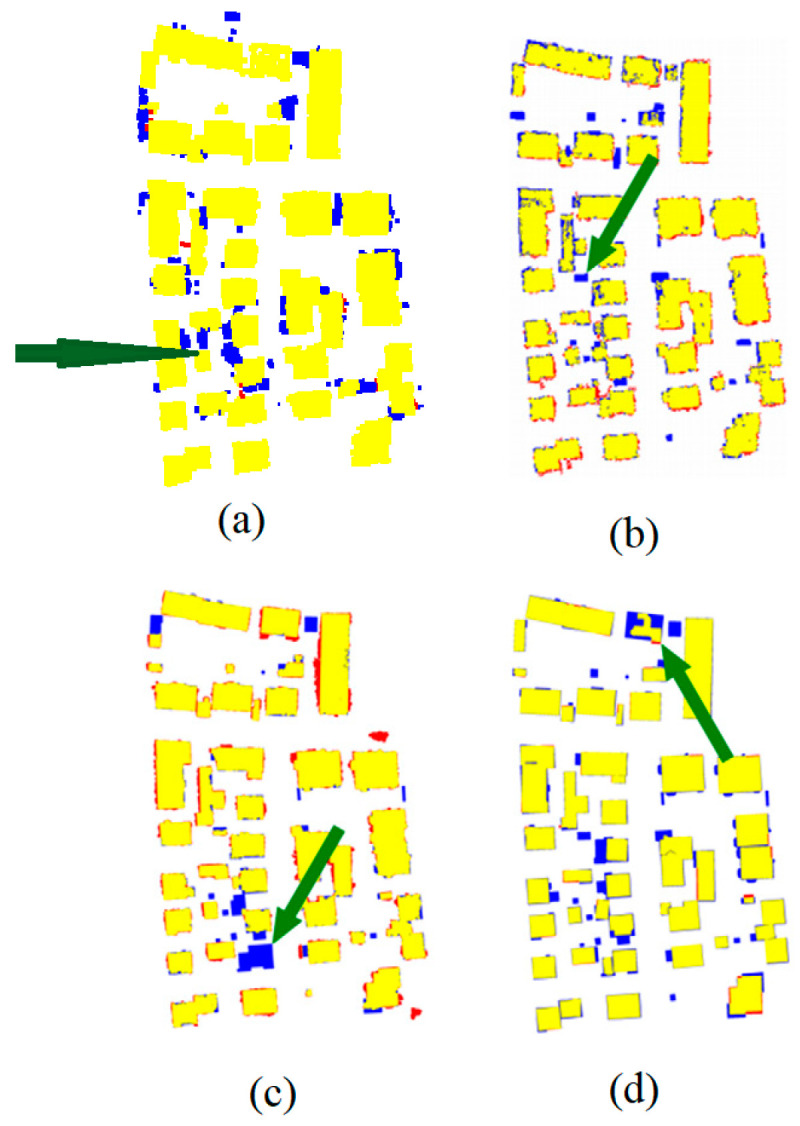
Comparison of second classification strategy result with previous studies using Vaihingen dataset; (**a**) Suggested approach; (**b**) Result obtained by Wei et al. [52]; (**c**) Result obtained by Rottensteiner et al. [53]; (**d**) Result obtained Dorninger et al. [54], Green arrows point to some considerable classification errors. (**b**–**d**) are taken from Rottensteiner et al. [51].

**Table 1 sensors-23-07360-t001:** Tested datasets.

	Brisbane 1	Brisbane 2	Brisbane 3	Vaihingen	Christchurch [39]
Area (ha)	100	7	98.7403	3	60
Number of points	12,217,017	35,000,000	10,683,855	230,303	12,029,508
Flight height (m)	2000	55	2060	500	-
Mean density (point/m^2^)	12.2	500	-	7.5	19
Theoretical density (point/m^2^)	2	250	2	4–6.7	-
Point accuracy (cm)	30–80	5–10	30–80	10–30	-

**Table 2 sensors-23-07360-t002:** Summary of feature values illustrated in Figure 3, Figure 4, Figure 5, Figure 6, Figure 7, Figure 8, Figure 9, Figure 10, Figure 11, Figure 12 and Figure 13.

Feature	VC	TC	BC	Acceptance
**Sum of eigenvalues**	0.1–0.52	0.02–0.12	0.4–0.73	No
**Omnivariance**	0.01–0.13	0.0025–0.016	0.03–0.1	No
**Eigenentropy**	0.3–0.75	0.1–0.3	0.7–0.8	Yes
**Planarity**	0.05–0.85	0–0.8	0.2–0.96	No
**Sphericity**	0–0.3	0–0.1	0–0.04	Yes
**Surface variation**	0–0.15	0–0.07	0–0.02	Yes
**Verticality**	0–1	0–0.05	0–0.1	Yes
**Anisotropy**	0.75–1	0.9–1	0.96–1	Yes
**Linearity**	0.1–0.95	0.2–1	0.04–0.8	No
**PCA1**	0.48–0.92	0.52–0.96	0.5–0.82	No
**PCA2**	0.06–0.46	0.02–0.46	0.18–0.49	No

VC: Vegetation Class; TC: Terrain Class; BC: Building Class.

**Table 3 sensors-23-07360-t003:** Percentage of class points in studied point clouds.

Point Cloud	Vegetation (%)	Buildings (%)	Terrain (%)	Sum (%)
Brisbane 1	21.78	10.1	63.29	95.17
Brisbane 2	26.34	31.34	32.24	89.92
Christchurch	8.36	35.2	41.9	85.46
Vaihingen	12.16	25.5	50.4	88.06

**Table 4 sensors-23-07360-t004:** Accuracy comparison of second classification strategy with previous studies.

Approach	Accuracy (%)
Wei et al. [52]	83.9 to 92.9
Rottensteiner et al. [53]	85.0 to 93.6
Dorninger et al. [54]	84.6 to 98.4
Strategy 2	98

**Table 5 sensors-23-07360-t005:** Extraction accuracy of various categories among first and second classification strategies.

Strategy	Buildings Class Accuracy (%)	Vegetation Class Accuracy (%)	Terrain Class Accuracy (%)
1	96.9	96.9	96.9
2	99	97	98

Strategy 1 is the classification of buildings, terrain, and vegetation; Strategy 2 is the classification of buildings and non-buildings. Then, the non-buildings class is classified into vegetation and terrain.

## Data Availability

Not applicable.

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
