# Peer review of "Contribution of Geometric Feature Analysis for Deep Learning Classification Algorithms of Urban LiDAR Data"

_sensors, 2023, doi:10.3390/s23177360_

Round 1

Reviewer 1 Report

This is a great study and the resulting article is well-written. My congratulations. I like the integraztion of geometric features into the AI approach.

Author Response

Thank you very much for your time and efforts in reviewing our manuscript. We appreciate your observations and comments.

Reviewer 2 Report

1. In the introduction, the author gives a general introduction to the application and processing methods of LiDAR.But there's no analysis about why we need to do geometric analysis. What is research significance and value of studying the contribution of the geometic feature analysis on the DL ?

2. In Related works, the author briefly introduces LiDAR data processing methods based on ML classification. Still, in the summary and introduction, only a simple listing and analysis of related classification methods and the advantages and disadvantages of each method are lacking. The logical structure of the whole presentation is not clear. What is the relationship between some of the research work listed in deep learning methods? The author needs to give an overarching analysis of the conclusion, and this conclusion is to justify your research work.

3. What is the data on which the geometric feature analysis is based in the fourth part? Is this part of data representative? At the same time, it is obvious from Table 2 that only three categories are highly distinguishable in the Omnivariance feature distribution, so the author needs to use more intuitive graphs or tables to more directly explain the reasons for selecting certain categories of features as model inputs.

4. The comparison method used by the author in ResultandAccuracy is only self-comparison, and no comparison experiments between the rule-based algorithm and machine learning algorithm mentioned in the abstract have been observed. At the same time, the overall experiment is too heavy to discuss the classification strategy, which has an impact on the final classification result, but it can not be a factor in evaluating the quality of a model, and the classification strategy should be bound to the actual demand. In addition, the second strategy gives priority to ensuring the extraction accuracy of buildings, and the author does not explain the variation of the accuracy of different categories among different strategies in the paper. The extraction accuracy of various categories should be presented in the form of a table to strengthen the discussion and analysis of the quantitative part.

5. The manuscript's formula numbering and map numbering formats are confused and should be adjusted uniformly.In addition, why does point cloud display need to be layered according to Z-coordinate values, and is this necessary to display experimental results?Authors are advised to color according to categories.

The English language is fine. Minor grammar problems should be revised.

Author Response

Thank you very much for taking the time to review our manuscript. We note from your feedback that you reviewed the paper very carefully line by line. We appreciate the great effort you put into this task. We feel extremely fortunate that our paper has received such attention from you. In fact, such high-quality reviewing reflects the outstanding quality of Sensors Journal. This is why we thank the journal editor for the careful selection of the review team. At this stage, we would like to confirm that the paper has been revised and edited by considering all your comments. All corrections are now highlighted in yellow. We hope that the revised version meets the required standard and is accepted for publication in this journal.

Reviewer 3 Report

The authors present a DL-based point cloud semantic classifier with a simple architecture compared to most recent methods. Instead they rely on a strong inductive bias by providing the network with a set of pre-selected geometric features. This goes somewhat against the grain, since most proposals let the network choose the relevant features. However, in spite of the simple network architecture (just a 10-layer MLP) the system seems to perform well, but because of this simplicity, I have some concerns about the ability to generalize of this solution. Therefore in order to convince the reader, the authors should test their trained classifier against one or more point clouds different from the four used for training, including as much variety as possible: different scale, orientation, scanning resolution and urban scenery.

Apart from this, the introduction to the method in the manuscript must be improved:

Paragraph in lines 200-206 is vague and poorly written. Please state more clearly the goals and contributions of the paper in the context of existing ML and DL-based methods for point cloud semantic classification. Since PointNet and PointNet++ were introduced in 2017, this has been a very active area of research, as evidenced in Section 2, so any new contribution must be put in context. As always in science and technology, presenting “just a new method” for an existing problem is not enough. 

Connected with the previous comment and before Section 3 (Input data) the method should be outlined to guide the reader from a general view of the method to its details. It is strange in section 208-209 to read “…to test the effectiveness of the suggested classification approach…” without having any clue of the suggested classification approach. Again in Section 4 the geometric features are described without explaining first why this is important for the method. We have to wait to Section 5 to have the full picture.

Some awkward sentences found, such as that in lines 200-201.

Author Response

(The authors gave the same response as above.)

Round 2

Reviewer 2 Report

The author has made better changes in accordance with the first review suggestions. In the revised version of Line 528-538, the author has added some discussion of the results of comparative experiments. However, this discussion needs to be supplemented by a more completed table showing accuracy comparison. Meanwhile, it is also necessary to add comparison classification results figures to show the experimental results to prove that the author's comparative experimental data is valid.

Author Response

Thank you very much for your time and efforts in reviewing our manuscript. We appreciate all your observations and comments. It is a great pleasure for us to consider all the comments and answer them as follows. Corresponding major changes are highlighted in the revised manuscripts.

Reviewer 3 Report

I appreciate the effort of the authors to improve the paper. Still, I believe that in order to be convincing the paper should support the claims about the proposed method with new experiments showing its performance with datasets different from those used during training. This has been included as future work. Anyway I can weak accept the paper in its present form. 

The English is correct. There are no major issues to read and understand the paper.

Author Response

Thank you very much for your time and efforts in reviewing our manuscript. We appreciate your observations and comments. It is a great pleasure for us to consider all the comments and answer them as follows. Corresponding major changes are highlighted in the revised manuscripts.
